# Stress distribution and surface shock wave of drop impact

Ting-Pi Sun[1], Franco Álvarez-Novoa[2], Klebbert Andrade[2], Pablo Gutiérrez [3], Leonardo Gordillo [2] & Xiang Cheng [1✉]

Drop impact causes severe surface erosion, dictating many important natural, environmental and engineering processes and calling for substantial prevention and preservation efforts. Nevertheless, despite extensive studies on the kinematic features of impacting drops over the last two decades, the dynamic process that leads to the drop-impact erosion is still far from clear. Here, we develop a method of high-speed stress microscopy, which measures the key dynamic properties of drop impact responsible for erosion, i.e., the shear stress and pressure distributions of impacting drops, with unprecedented spatiotemporal resolutions. Our experiments reveal the fast propagation of self-similar noncentral stress maxima underneath impacting drops and quantify the shear force on impacted substrates. Moreover, we examine the deformation of elastic substrates under impact and uncover impact-induced surface shock waves. Our study opens the door for quantitative measurements of the impact stress of liquid drops and sheds light on the origin of low-speed drop-impact erosion.

[1] Department of Chemical Engineering and Materials Science, University of Minnesota, Minneapolis, MN 55455, USA. [2] Departamento de Física, Facultad de Ciencia, Universidad de Santiago de Chile (USACH), Santiago, Chile. [3] Instituto de Ciencias de la Ingeniería, Universidad de O'Higgins, Rancagua, Chile.
✉email: xcheng@umn.edu

Laozi, the ancient Chinese philosopher in the fifth century BCE, has long noticed that water, although the softest and weakest material known in his time, is effectual in eroding hard substances[1]. Although Laozi used this attribute of water only as a metaphor to extol the virtues of humbleness, flexibility, and persistence, it was a physically nontrivial observation that dripping water drops, with zero shear modulus and readily deformable, can erode hard solid substrates with finite yield stresses (Fig. 1a, b)[2]. Beyond its philosophical meaning, erosion by drop impact is relevant for a wide range of natural, environmental, and engineering processes including soil erosion[3–7], preservation of heritage sites[8], wear of wind and steam turbine blades[9], and cleaning and peening of solid materials (e.g. silicon wafers)[10–12]. While the impact damage on solid substrates caused by high-speed compressible liquid drops with the impact velocity on the order of a few hundred meters per second has been well explored[13,14], our understanding of the impact erosion of low-speed incompressible drops relevant to most natural and engineering processes is still rudimentary. Why is low-speed drop impact erosive? What are the key dynamic features of drop impact that lead to its unexpected ability in erosion? The answer to these questions would provide not only the testimony to the wisdom of the ancient philosopher but also fundamental insights into the early-time dynamics of drop impact.

Due to its ubiquity in nature and industry, drop impact has evoked long-lasting research interests since the early study of Worthington almost one and a half centuries ago[15]. Particularly, significant progress has been made over the last two decades in understanding drop impact thanks to the fast advance of high-speed photography techniques. Nevertheless, limited by direct imaging, most current studies focused on the kinematics of impacting drops such as the splashing threshold, the maximum spreading diameter and contact time, and the formation of cushioning air layers underneath drops (see refs. [16,17] and references therein). Very few experiments have been performed probing the dynamic properties of drop impact that are directly responsible for drop-impact erosion[18]. As an important dynamic factor, the impact force of liquid drops has begun to attract attention in recent years[18–26]. However, it becomes clear from these recent studies that the unusual ability of an impacting drop in erosion cannot stem from its impact force alone, as the maximum impact force induced by a millimetric water drop falling near its terminal velocity is very weak, more than an order of magnitude smaller than that generated by the impact of a solid sphere of similar size, density, and velocity[23]. The average impact pressure of the drop is even smaller due to the large contact area formed by the spreading drop[2]. Thus, instead of impact force or average impact stresses, the erosion ability of drop impact must originate from the unique spatiotemporal structure of its impact stresses as well as the dynamic response of impacted substrates under such stresses. Nevertheless, quantitative measurements on the stress distributions of drop impact have not been achieved heretofore due to the limitation of existing experimental tools. Recent attempts using an array of miniature force sensors based on microelectromechanical systems (MEMS) provided only the coarse-grained dynamics of impact pressure without a sufficient spatial resolution and failed to measure the shear stress distribution of impacting drops[27,28]. Here, we develop an experimental method—high-speed stress microscopy—to measure the impact stress of drop impact on solid elastic substrates. The method integrates the imaging techniques of traction force microscopy[29], laser-sheet microscopy, and high-speed photography, which allows us to map the temporal evolution of the pressure and shear stress distributions underneath millimeter-sized drops in fast impact events with unprecedented spatiotemporal resolutions.

## Results

**High-speed stress microscopy.** We embed low-concentration (0.23% v/v) fluorescent polystyrene particles of diameter 30 μm in a cross-linked polydimethylsiloxane (PDMS) gel as tracers to track the deformation of the gel under impact. The PDMS gel surface is hydrophobic with the water contact angle ~90°. Young's modulus of the gel is fixed at $E = 100$ kPa in our experiments, although gels with $E$ up to 420 kPa and with hydrophilic surfaces have also been tested (Methods). A thin laser sheet of 30-μm thickness illuminates the gel from a side and excites the fluorescent tracers within the sheet (Fig. 2a). The sheet is finely adjusted to be normal to the impacted surface and to pass through the center of impacting drops. A high-speed camera focusing on the sheet images the motion of tracers at 40,000 frames per second.

We track the displacements of the tracers from the high-speed video using digital image correlation (DIC). An interrogation window of 384 μm by 384 μm with 70% overlap is adopted in DIC, which gives a spatial resolution of 115 μm. The temporal resolution is 0.025 ms, set by the frame rate of high-speed photography. To reduce measurement errors, we first average the cross-correlation fields of DIC from five repeated impacts on the same gel at the same impact location. The surface is fully dried between two consecutive experimental runs. The stress measurements are the outcome of a further average over three different averaged displacement fields for impacts on different gels or impacts on the same gels at different impact locations. Thus, one

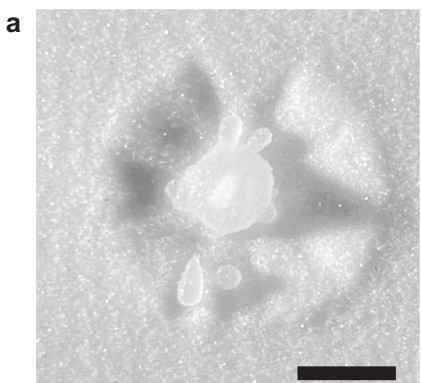

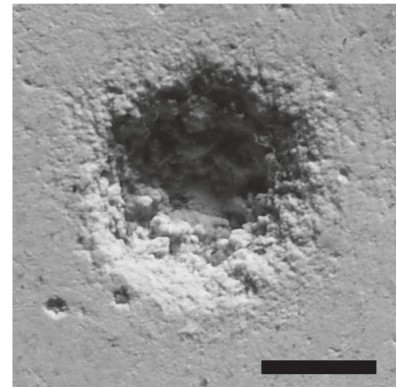

**Fig. 1 Erosion by dripping drops. a** Erosion crater by drop impact on a granular medium made of 90 μm glass beads. The crater is created by the impact of a single water drop of diameter $D = 3$ mm at impact velocity $U = 2.97$ m/s. **b** Erosion crater by drop impact on a plaster slab. The crater is created by 2500 repeated impacts of water drops of $D = 3$ mm at $U = 2.6$ m/s. Scale bars: 5 mm.

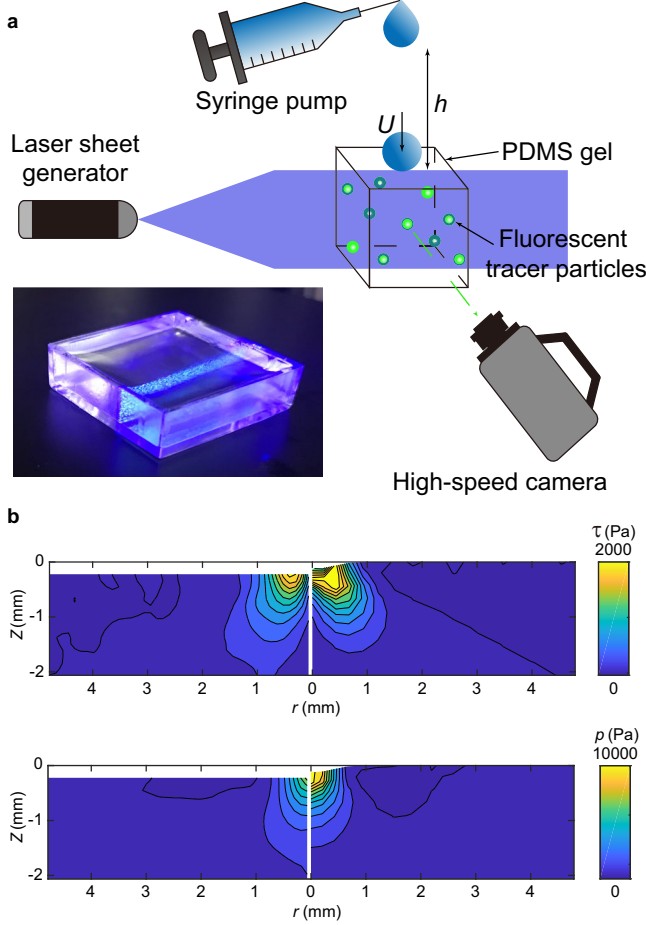

**Fig. 2 High-speed stress microscopy. a** A schematic showing the principle of high-speed stress microscopy. A drop falls from a height $h$ and impacts onto the surface of a particle-embedded PDMS gel at an impact velocity $U$. Lower left inset: An image of the PDMS gel embedded with fluorescent particles under the illumination of the laser sheet. **b** Comparison of the shear stress (top) and the pressure (bottom) induced by the impact of a steel sphere obtained from high-speed stress microscopy (left) and from finite element simulations (right). The diameter and the impact velocity of the steel sphere are $D = 3.16$ mm and $U = 0.49$ m/s, respectively. The stresses are measured at time $t = 0.25$ ms after the instant when the sphere first touches the solid surface.

data point represents the average result of total 15 different experimental runs.

Stress fields depend on strain fields, which are the derivative of the displacement fields obtained from DIC. A smoothing procedure is necessary in order to reduce the noise of differentiation. We implement the moving least squares (MLS) interpolation method to obtain a continuously differentiable displacement field $u(r, z) = [u_r(r, z), u_z(r, z)]$ from the discrete displacement field of DIC[30]. A third-order polynomial basis is adopted in the interpolation (Supplementary Information (SI) Section 1).

When the deformation is small, the strain components in cylindrical coordinates follow:

$$\varepsilon_{rr} = \frac{\partial u_r}{\partial r}, \varepsilon_{zz} = \frac{\partial u_z}{\partial z}, \varepsilon_{\theta\theta} = \frac{u_r}{r}, \varepsilon_{rz} = \frac{1}{2}\left[\frac{\partial u_r}{\partial z} + \frac{\partial u_z}{\partial r}\right], \quad (1)$$

where we take the advantage of the cylindrical symmetry of the drop-impact geometry. By assuming the PDMS gels are isotropic

and linear following the generalized Hooke's law at small strains, we calculate the stress fields using the linear stress–strain relation:

$$\sigma_{ij} = \lambda\varepsilon_b\delta_{ij} + 2G\varepsilon_{ij}, \quad (2)$$

where $\lambda = E\nu/[(1 + \nu)(1 - 2\nu)]$ is the Lamé coefficient, $G = E/[2(1 + \nu)]$ is the shear modulus, $\delta_{ij}$ is the Kronecker delta and $\varepsilon_b \equiv \varepsilon_{zz} + \varepsilon_{rr} + \varepsilon_{\theta\theta}$ is the bulk strain. $\sigma_{rz}$ gives the shear stress $\tau$, whereas $\sigma_{zz}$ gives the pressure $p$. PDMS gels are nearly incompressible with Poisson's ratio $\nu$ close to 0.5, which result in a large $\lambda$. But the bulk strain $\varepsilon_b$ is close to 0 in this limit. Therefore, the impact pressure cannot be accurately determined from the product of $\lambda\varepsilon_b$ in Eq. (2). Instead, we adopt a quasi-steady state assumption to calculate the pressure[30], a procedure detailed further in SI Section 2. We have verified the assumption by comparing the inertial force and the elastic force in the impact process (SI Section 2) and by comparing experimental and numerical results on the impact pressure of solid-sphere impact (see below). The shear stress, on the other hand, is not affected by the nearly incompressible condition. The surface stresses and displacements are finally obtained at a location slightly below the original impacted surface (Methods).

As a calibration and the basis of comparison, we first measure the pressure and shear stress induced by the impact of a solid steel sphere of diameter $D = 3.16$ mm at impact velocity $U = 0.49$ m/s and compare the results with those from finite element simulations (Methods). Experimental measurements agree well with the numerical results, validating the accuracy of high-speed stress microscopy (Fig. 2b).

For drop impact, our drops are made of an aqueous solution of sodium iodide (60% w/w), which has a density $\rho = 2.2$ g/ml and a viscosity $\eta = 1.12$ mPa s[31]. The surface tension of the solution $\sigma \approx 81.3$ mN/m from pendant-drop tensiometry[32], which is slightly larger than that of water. We fix the diameter $D$ and impact velocity $U$ of drops in our experiments. Drops of $D = 3.49$ mm impact normally on the surface of PDMS gels at $U = 2.97$ m/s, yielding a Reynolds number $Re = \rho UD/\eta = 20,360$ and a Weber number $We = \rho DU^2/\sigma = 833$. Thus, the drop impact is dominated by fluid inertia at early times. We focus on drop impact at early times below, when the shear stress and pressure of impacting drops are high for strong erosion. Positions and times are reported in dimensionless forms using $D$ and $D/U$ as the corresponding length and time scale, respectively.

**Impact shear stress.** Surface erosion is the direct consequence of the shear stress of drop impact. Figure 3a, b compares the temporal evolution of the shear stress of solid-sphere impact and liquid-drop impact. Upon the impact, spatially non-uniform shear stresses quickly develop in both cases. However, while the position of the maximum shear stress of solid-sphere impact is stationary near the impact axis at $r = 0.095$, the maximum shear stress of drop impact propagates radially with the spreading drop. The kymographs of the surface shear stress, $\tau(r, z = 0, t)$, of the two impact processes are shown in Fig. 3c, d, highlighting further the fast propagation of the maximum shear stress of drop impact.

To understand the origin of the maximum shear stress of drop impact, we correlate the position of the maximum shear stress $r_s$ with the shape of impacting drops (Fig. 4a). Two kinematic features are analyzed: the tip of the expanding lamella $r_{lm}$ and the turning point $r_t$, where the drop body connects to the root of the lamella (Fig. 4a inset). It should be emphasized that $r_t$ is not the contact line of the drop. The ejection of the lamella occurs around $t \approx We^{-2/3} = 0.0104$[33], which is shorter than the temporal resolution of our experiments. While $r_{lm}$ moves fastest, $r_s$ follows closely behind $r_t$. Thus, the maximum shear stress arises from the strong velocity gradient near the turning point,

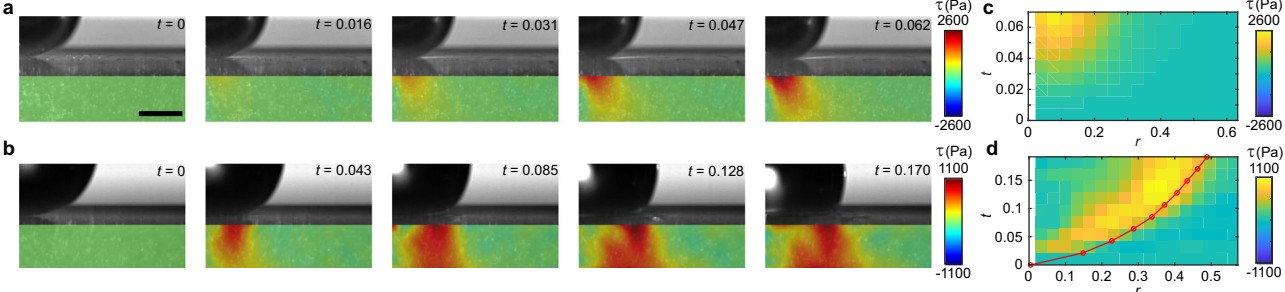

**Fig. 3 Shear stress distribution of solid-sphere impact and drop impact. a, c** The temporal evolution of the shear stress $\tau(r, z, t)$ and the kymograph of the surface shear stress $\tau(r, z = 0, t)$ of solid-sphere impact. The diameter and the impact velocity of the steel sphere are $D = 3.16$ mm and $U = 0.49$ m/s, respectively. **b, d** The temporal evolution of $\tau(r, z, t)$ and the kymograph of $\tau(r, z = 0, t)$ of drop impact. $D = 3.49$ mm and $U = 2.97$ m/s for the liquid drop. The red line in (**d**) indicates the position of the turning point. $t = 0$ corresponds to the instant when the impactors first touch the surface of the PDMS gels. Scale bar in (**a**) is 1 mm.

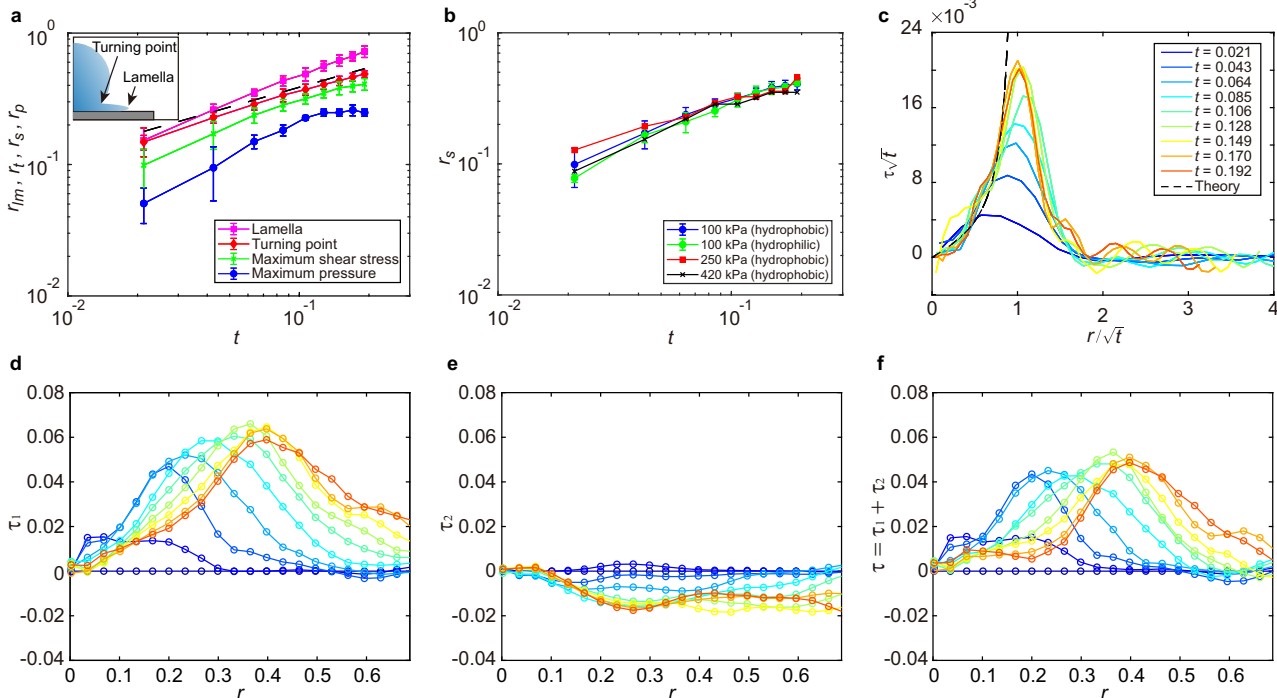

**Fig. 4 Surface shear stress of drop impact. a** The position of the lamellar tip $r_{lm}$, the turning point $r_t$, the maximum shear stress $r_s$ and the maximum pressure $r_p$ as a function of time $t$. The dashed line indicates $r = \sqrt{6t}/2$. Upper inset: Definition of the kinematic features of an impacting drop. **b** The position of the maximum shear stress of drop impact $r_s(t)$ on substrates of different Young's moduli and wettability. The error bars are the standard deviation of 15 experimental runs. **c** The rescaled surface shear stress $\tau/\sqrt{t}$ as a function of the rescaled radial position $r/\sqrt{t}$. The dashed line is the prediction of Eq. (3) with the modified scaling function $f(x)$. **d, e** Two components of the surface shear stress: $\tau_1 = G(\partial u_z/\partial r)|_{z=0}$ and $\tau_2 = G(\partial u_r/\partial z)|_{z=0}$. **f** The total surface shear stress $\tau = \tau_1 + \tau_2$. The shear stresses in (**c–f**) are nondimensionalized by $\rho U^2$ and share the same color code for time, as indicated in (**c**). The diameter and the impact velocity of the liquid drop are $D = 3.49$ mm and $U = 2.97$ m/s.

where the flow changes rapidly from the downward vertical direction (the $-z$ direction) within the drop body to the horizontal radial direction (the $r$ direction) inside the narrow lamella[34]. Quantitatively, $r_t(t)$ follows the well-known square-root scaling $r_t(t) = \sqrt{6t}/2 \approx 1.22\sqrt{t}$ established by many previous experiments[23,33,35–39]. In comparison, $r_s(t)$ also shows a square-root scaling with a slightly smaller prefactor $r_s(t) \approx \sqrt{t}$. The positions of the maximum shear stress and the turning point are independent of the wettability or Young's modulus of PDMS gels (Fig. 4b).

Philippi et al.[40] proposed that the shear stress of incompressible drops on infinitely rigid substrates possesses a self-similar

dynamic structure when $t \to 0^+$,

$$\tau(r, z = 0, t) = 2\sqrt{\frac{6}{\pi^3 Re}} \frac{1}{\sqrt{t}} f\left(\frac{r}{\sqrt{t}}\right) \quad \text{for} \quad r \le r_t(t), \quad (3)$$

where the scaling function $f(x) = x/(3 - 2x^2)$ dictates a finite-time singularity at the turning point $r_t(t)$. Here, $\tau(r, t)$ is non-dimensionalized by the inertial pressure $\rho U^2$. Note that the much stronger water-hammer pressure $\rho Uc$ associated with the compression wave occurs on the time scale of a few nanoseconds, which is too short to be relevant in our current experiments[7,20,23]. Here, $c$ is the speed of sound in liquid. Inspired by the self-similar hypothesis, we plot $\tau\sqrt{t}$ versus $r/\sqrt{t}$ of our experimental results (Fig. 4c),

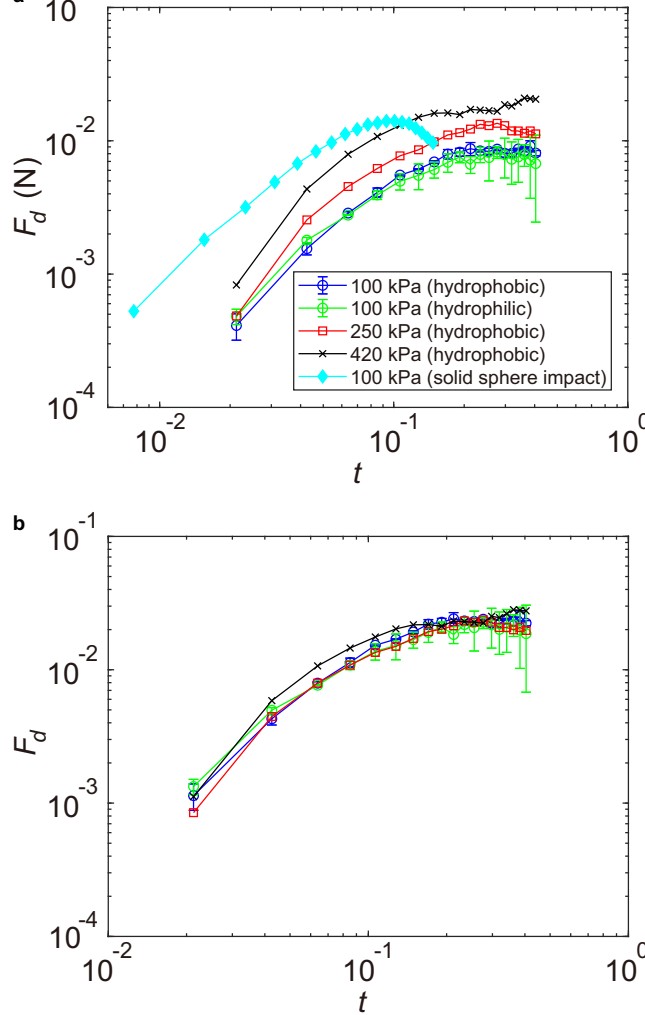

**a**

**b**

**Fig. 5 Shear force of drop impact. a** The shear force $F_d(t)$ of drop impact and solid-sphere impact on the impacted surface of different Young's moduli and wettability. The leftmost curve is for solid-sphere impact and the rest are for drop impact as indicated in the legend. **b** The dimensionless shear force $F_d$ of drop impact scaled by $E^{1/2}(\rho U^2)^{1/2}D^2$, as suggested by the theoretical analysis of drop impact on elastic substrates (SI Section 3). The diameter and the impact velocity of the liquid drop are $D = 3.49$ mm and $U = 2.97$ m/s. $D = 3.16$ mm and $U = 0.49$ m/s for the steel sphere.

which shows a good collapse at a small $r$ away from the singular region. With a modified scaling function $f(x) = x/(1 - x^2)$ to count the different temporal scalings of $r_t$ and $r_s$, the collapsed data quantitatively agrees with Eq. (3) (the dashed line in Fig. 4c). Thus, our study provides experimental evidence on the propagation of shear stress of drop impact and demonstrates the self-similar structure of shear stress at early times.

Despite the general agreement with Eq. (3) at $r < r_t$, our experiments also reveal the unique features of drop impact on elastic deformable substrates, absent in the theoretical consideration of drop impact on infinitely rigid substrates. The shear stress on the surface of an elastic substrate is given by $\tau = G(\partial u_r/\partial z + \partial u_z/\partial r)$ (Eqs. (1) and (2)), where $G$ is the shear modulus of the substrate and $u_r$ and $u_z$ are the radial and vertical displacement of the substrate surface. We find $|\partial u_z/\partial r| > |\partial u_r/\partial z|$ (Fig. 4d–f), suggesting the dominant role of the vertical velocity of the impacting drop at the contact surface $v_z(r, z = 0)$ on the shear stress. Note that $u_z(r, z = 0, t) = \int_0^t v_z(r, z = 0, t)dt$.

For drop impact on infinitely rigid substrates, $v_z(r, z = 0, t) = 0$ because of the no-penetration boundary condition, which inevitably gives $\partial u_z/\partial r = 0$. Instead, the shear stress of drop impact on infinitely rigid substrates arises from the gradient of the radial velocity, $\partial v_r/\partial z$, within the boundary layer near the contact surface. As $u_z(r, z = 0)$ is mainly determined by the pressure distribution on the contact surface at high $Re$, this finding illustrates the intrinsic coupling between the impact pressure and shear stress of drop impact on elastic substrates.

The effect of the finite stiffness of the impacted substrate also manifests in the shear force of impacting drops. By integrating the shear stress over the contact area, we obtain the shear force, $F_d(t) = 2\pi \int_0^{r_{lm}} \tau(r, z = 0, t)rdr$, which quantifies the total erosion strength of drop impact. Although $F_d(t)$ is independent of the wettability of the impacted surface, it increases with Young's modulus following a scaling $F_d \sim \sqrt{E}$ within the range of our experiments (Fig. 5a, b). Thanks to the spreading of the maximum shear stress, drop impact and solid-sphere impact show comparable peak shear forces under similar impact conditions (Fig. 5a).

**Impact pressure and surface shock wave.** Although subject to larger experimental errors due to the nearly incompressibility of PDMS, the pressure (i.e. normal stress) distribution underneath impacting drops $p(r)$ can be also measured by high-speed stress microscopy (SI Section 2). Similar to the shear stress, we observe a non-central pressure maximum propagating radially with the spreading drop (Fig. 6b, d). The dynamics are again in sharp contrast to the pressure of solid-sphere impact, where the maximum impact pressure is fixed at the impact axis $r = 0$ (Fig. 6a, c). The existence of the propagating non-central pressure maximum has been predicted by several theories and simulations of drop impact[38,40–43]. Nevertheless, to the best of our knowledge, such a counter-intuitive prediction has not been directly verified in experiments heretofore. While our measurements qualitatively confirm the prediction, we find that the maximum pressure falls behind the maximum shear stress (Fig. 4a), a feature unexpected from drop impact on infinitely rigid substrates[40].

More interestingly, a negative pressure emerges in front of the turning point $r_t$ at $t_c \approx 0.106$ (Fig. 6b, d). In the meantime, we also observe the propagation of surface disturbance on the gel surface away from the stress maxima above $t_c$ (Fig. 7a). Both suggest the formation of a surface acoustic wave—the classic Rayleigh wave—in the gel. Since the speed of the turning point $V_t(t) = dr_t/dt = \sqrt{6}/(4\sqrt{t})$ increases with decreasing $t$, the stress maxima associated with the turning point spread supersonically at early times (Fig. 7b). Thus, a shock front forms near $r_t$ on the impacted surface when $t < t_c$. The Rayleigh wave finally overtakes the turning point and is released in front of the spreading drop in an explosion-like process above $t_c$, giving rise to the negative pressure and the propagation of surface disturbance. Based on the above picture, the speed of the surface wave can be estimated as $V_t(t_c) = \sqrt{6}/(4\sqrt{t_c}) = 1.88$, which quantitatively matches the speed of the Rayleigh wave[44]

$$V_R = \frac{1}{M}\left[\sqrt{\frac{1}{2(1 + \nu)}}\frac{0.862 + 1.14\nu}{1 + \nu}\right] = 1.89. \quad (4)$$

Here, the Mach number $M \equiv U\sqrt{\rho_s/E} = 0.292$ with $\rho_s = 0.965$ g/cm³ and $\nu = 0.49$ is the density and Poisson's ratio of PDMS.

Encouraged by the quantitative agreement between Eq. (3) and experiments, we couple the theoretical impact pressure and shear stress of incompressible drops on infinitely rigid surfaces with the Navier-Lamé equation of semi-infinite elastic media (SI Section 3). The dimensional analysis of the governing equation and the boundary conditions suggests that the shear force should scale as

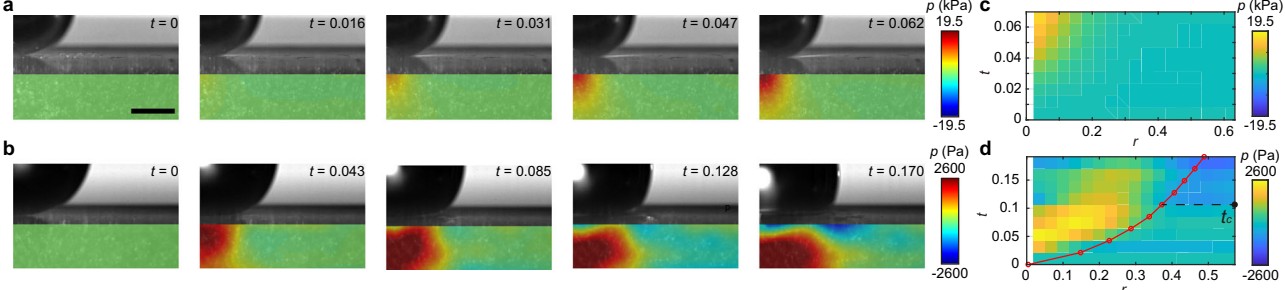

**Fig. 6 Pressure distribution of solid-sphere impact and drop impact. a, c** The temporal evolution of the pressure $p(r, z, t)$ and the kymograph of the surface pressure $p(r, z = 0, t)$ of solid-sphere impact. The diameter and the impact velocity of the steel sphere are $D = 3.16$ mm and $U = 0.49$ m/s. **b, d** The temporal evolution of $p(r, z, t)$ and the kymograph of $p(r, z = 0, t)$ of drop impact. $D = 3.49$ mm and $U = 2.97$ m/s. The red line in **d** indicates the position of the turning point. The time when the negative pressure emerges $t_c = 0.106$ is also indicated. Scale bar in (**a**) is 1 mm. Note that the pressure scale is kPa for the solid-sphere impact and Pa for the drop impact.

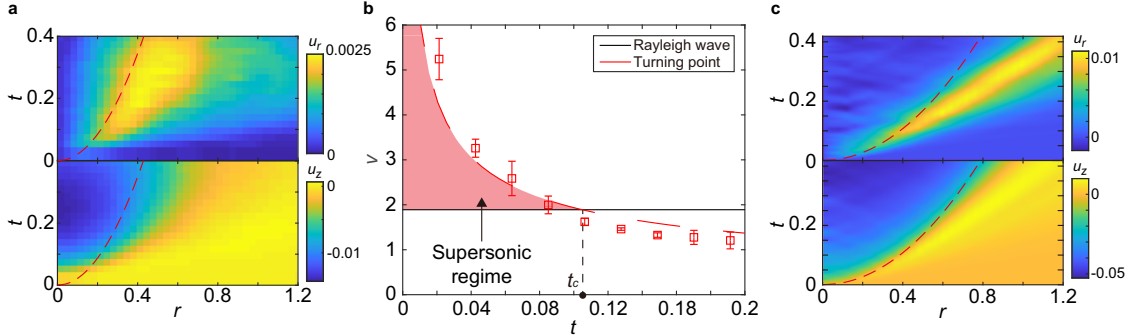

**Fig. 7 Surface shock wave of drop impact. a** The kymograph of the radial displacement of gel surface $u_r(r, t)$ (top) and the vertical displacement of gel surface $u_z(r, t)$ (bottom) induced by drop impact. The diameter and the impact velocity of the drop are $D = 3.49$ mm and $U = 2.97$ m/s. The displacements are nondimensionalized by $D$. **b** Competition between the speed of the turning point $V_t$ and the speed of the Rayleigh wave $V_R$. The supersonic regime before $t_c$, where $V_t > V_R$, is indicated. Symbols are from experiments, where the error bars are the standard deviation of 15 experimental runs. The red dashed line shows $V_t = \sqrt{6}/(4\sqrt{t})$. **c** The numerical solution of $u_r(r, t)$ (top) and $u_z(r, t)$ (bottom) induced by drop impact. The red dashed lines in (**a**) and (**c**) indicate the position of the maximum pressure that drives the Rayleigh wave. Note that the maximum pressure propagates with the turning point in the simulation, which is faster than that observed in experiments (Fig. 6d). See SI Section 3 for more details about the numerical solution.

$F_d \sim E^{1/2}(\rho U^2)^{1/2} D^2$, agreeing with our measurements at different $E$ (Fig. 5b). Moreover, the numerical solution of the coupled equations qualitatively reproduces the formation of the shock-induced Rayleigh wave of drop impact, where a sharp surface wave with a well-defined peak emerges at $t_c \approx 0.1$ and propagates with $V_R$ (Fig. 7c). The strong and sharp surface wave is produced by the mechanical resonance occurring when the speed of the stress maxima approaches the speed of the Rayleigh wave near $t_c$. Such a resonant phenomenon does not exist for solid-sphere impact with stationary stress maxima. As a result, the surface Rayleigh wave of solid-sphere impact is more diffusive (Supplementary Fig. 2).

## Discussion

Taken together, our high-speed stress microscopy reveals three unique dynamic features of drop impact. These features contribute to the ability of drop impact to erode solid surfaces and result in the distinct nature of drop-impact erosion, which is qualitatively different from that of solid-sphere-impact. (i) The spatiotemporal stress distributions of impacting drops are highly non-uniform. The radially propagating stress maxima simultaneously press and scrub impacted substrates, leading to a large erosion area and a high shear force. (ii) Because of the fast speed of the turning point at short times, a shock wave forms on impacted substrates, which substantially increases the erosion strength. Each impacting drop behaves like a tiny bomb, releasing its kinetic impacting energy explosively. (iii) A sharp shock-

induced surface wave finally emerges from the explosion process. The resulting decompression wave weakens the cohesion of surface materials before the arrival of the shear stress maximum.

Can drop impact induce surface shock waves on substrates stiffer than the PDMS gels used in our experiments? Although the turning point $r_t(t)$ exhibits a square-root temporal scaling $\sqrt{6t}/2$, which suggests a divergent speed as $t \to 0^+$ and therefore supports surface shock waves on infinitely rigid substrates, such a singular behavior is regularized at small times due to the development of compression waves and/or the effect of air cushioning. Nevertheless, even at the shortest time of our experiments of $t = 0.021$, the square-root scaling still holds well (Fig. 4a). Previous experiments on the kinematics of drop impact have shown the scaling at even smaller times down to $t \approx 5 \times 10^{-4}$ before the ejection of lamella (Fig. 7 in ref. [39]), where the turning point is coincident with the apparent contact line of impacting drops. The speed of the turning point at this time is $V_t = \sqrt{6}/(4\sqrt{t}) = 27.4$. If we set the $V_t$ to be the speed of the Rayleigh wave (Eq. (4)), the upper limit of Young's modulus of elastic substrates where we still expect to observe surface shock waves is $E \approx 20$ MPa. Here, $\nu = 0.49$ and $\rho_s = 1000$ kg/m$^3$ are taken in the estimate. Compression waves develop within impacting drops at very early times $t \approx U^2/c^2 \approx 4 \times 10^{-6}$ [20]. The speed of the turning point at such a small time scale is $V_t = 300$, which gives a theoretical upper limit of Young's modulus $E \approx 3$ GPa, approaching the modulus of sedimentary rocks.

The effect of air cushioning is more complicated. By preventing the on-axis contact, air cushioning eliminates both the compression

waves and the divergent speed of the turning point as $t \to 0^{+}$[42,43]. Nevertheless, as discussed above, numerous experiments have repeatedly confirmed the square-root temporal scaling of the turning point in the ambient air at small time scales[23,33,35–39], supporting the fast propagation of the turning point in air at short times. More importantly, theories and simulations have both shown that air cushioning does not annihilate the fast propagation of the non-central pressure peak[40,42,43]. Particularly, the underlying air layer induces a micron or sub-micron dimple-like deformation bounded by a kink structure at the bottom of an impacting drop[42,45]. The pressure reaches the maximum underneath the kink[42,43], which propagates radially outwards at a speed 50 times higher than the impact velocity at early times of $t \sim 10^{-4}$[42,45] Hence, surface shock waves induced by the fast propagation of the pressure maximum should persist in the presence of air cushioning on substrates of Young's modulus at least up to ~70 MPa. It is an open question if surface shock waves can sustain on even stiffer substrates. The radius and the thickness of the air dimple decrease at reduced ambient pressure[46], which eventually leads to the vanishing of air cushioning and the recovery of the singular dynamics of the contact line at short times.

Lastly, it is worth noting that shock propagation within impacting drops and along impacted surfaces has been investigated theoretically for compressible drops with the liquid Mach number $U/c \sim O(1)$[13,14]. Nevertheless, the shock process of incompressible drops with $U/c \ll 1$ that are relevant to most natural and industrial processes has not been discussed heretofore. The application of high-speed stress microscopy in our study demonstrates its great potential to measure the impact stress of low-speed incompressible drops in more diverse situations such as drop impact on patterned substrates, at reduced ambient pressure and with non-Newtonian drops[16,17].

## Methods

**Preparation and characterization of PDMS substrates**. PDMS elastomers were prepared from two-part Sylgard 184 silicone elastomer kits (Dow Corning). To adjust Young's modulus, $E$, of the cured PDMS gels, we mixed siloxane monomers with crosslinkers at a controlled mass ratio. In most experiments reported in this paper, we used a monomer-to-crosslinker mass ratio of 30:1, which yielded PDMS gels of $E = 100$ kPa. PDMS gels of $E = 250$ kPa (mass ratio 20:1) and 420 kPa (mass ratio 18:1) have also been tested to assess the effect of gel stiffness on the impact pressure and shear stress (Figs. 4b and 5). Fluorescent polystyrene (PS) particles of diameter 30 μm (ThermoFisher) were mixed into the two-part PDMS mixtures at a volume fraction of 0.23% before curing. The PDMS-particle mixture was finally vacuumed to remove air bubbles and placed in an oven at 90 °C overnight for curing. The fully cured gels have a fixed thickness of 6.5 mm and an area of 24 mm by 24 mm, which is much larger than the maximum spreading area of impacting drops.

We modified the wettability of the surface of the PDMS gels via plasma modification. Without the modification, the untreated PDMS surfaces are hydrophobic with a contact angle of around 90°. After the modification, the surfaces become hydrophilic with a contact angle of less than 10°. Untreated hydrophobic gels were used in all the experiments reported in this study unless stated otherwise.

We measured Young's modulus of the cured gels, $E$, by surface indentation. Specifically, we used a steel ball of radius $R = 2.5$ mm as an indenter. Additional weight was also applied on the top of the sphere to indent the gels. The indentation length $d$ at a given indentation force $F$ was measured from the side view. $E$ was then calculated via the Hertzian contact law, $E = 9F/(16R^{1/2}d^{3/2})$[47], where we assume the deformation of the steel ball is negligible and Poisson's ratio of the gel is $v \approx 0.5$. We also estimated $E$ independently by matching the experimental stress distributions of solid-sphere impact with those from finite-element simulations. $E$ measured using these two different methods match well with at most 20% difference from different experimental runs. The values also agree quantitatively with the previous studies[48]. Poisson's ratio of the gels, $v$, is more difficult to assess accurately. We used $v = 0.49$ based on previous studies of the mechanical properties of PDMS gels under small strains[49].

**Surface stresses and displacements**. Two approximations were taken to estimate the surface stresses and displacements. First, we defined $z = 0$ as the horizontal plane through the lowest contact point between impactors and impacted substrates. As the maximum deformation of the PDMS gels under drop impact was only 92 μm comparable to the spatial resolution of DIC at 115 μm, we did not expect the approximation leads to large experimental errors. Second, to avoid any potential

boundary-induced artifacts in DIC, we took the stresses and displacements at $z = -315$ μm below the surface as the surface stresses and displacements. We verified that the stress distributions at shallower heights show quantitatively similar features although noisier. Particularly, the locations of the maximum shear stress and pressure do not vary with $z$ over this range. As a calibration, the procedure yielded a good approximation of the surface stress of solid-sphere impact at a larger maximum deformation of 192 μm (Fig. 2b).

**Finite element analysis**. To test the accuracy of high-speed stress microscopy (Fig. 2b), we used the commercial finite-element software ABAQUS to simulate solid-sphere impact, which has a shorter time scale than drop impact under similar impact conditions (Fig. 5a). The impact geometry was axisymmetric. The element shape for the meshing of the impacted surface was quadrilateral with adjustable sizes. Near the impact point, the element size was 0.2 mm. The diameter and the impact velocity of the impacting solid sphere were 3.16 mm and 0.49 m/s, matching the impact condition of the experiments. Since Young's modulus of the steel sphere is much larger than that of the impacted PDMS substrate, the sphere is assumed to be rigid in the simulation. The substrate is isotropic and linearly elastic. While Poisson's ratio of the substrate was fixed at 0.49, Young's modulus of the substrate was chosen to match the outcome of the experiments. Thus, the simulation allowed us to assess Young's modulus of the PDMS gels, independent of the indentation measurements discussed above. The two methods yield quantitatively similar results, agreeing well with the literature value.

To numerically solve the coupled equations of the impact stress of impacting drops and the deformation of elastic media (Fig. 7c), we used the partial differential equation toolbox in Matlab. Specifically, we adopted the transient axisymmetric geometry in the structural mechanics analysis of the toolbox. The pressure and shear stress distributions of incompressible drops on infinitely rigid substrates (Eqs. (14) and (15) in SI) were assigned as the boundary loads on an elastic medium, which has Young's modulus and Poisson's ratio matching those of experiments. We then numerically calculated the radial and vertical displacements of the surface of the elastic medium as a function of time (Fig. 7c). The dimension and the grid size of the medium are chosen so that the results are convergent, independent of these parameters. A detailed discussion of the coupled differential equations and their numerical solutions for both drop impact and solid-sphere impact can be found in SI Sec. 3.

## Data availability

The data supporting the main findings of this study are available in the paper and its Supplementary Information. Any additional data can be available from the corresponding author upon request.

## Code availability

The codes that support the findings of this study are available from the corresponding author upon reasonable request.

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

## Acknowledgements

We thank Ben Druecke, Tai-yin Chiu, and Grace Lee for help with experiments and data analysis and Michelle Driscoll for fruitful discussions. This research is supported by the US National Science Foundation CBET-2017071 and 2002817 and ACS Petroleum Research Fund 60668-ND9. T.-P.S. acknowledges the partial financial support of the PPG fellowship via UMN IPRIME and the Government Scholarship to Study Abroad from Taiwan. F.A.-N., K.A., L.G., and P.G. acknowledge the financial support of the grants ANID/CONICYT Fondecyt Iniciación No. 11170700 and 11191106.

## Author contributions

T.-P.S. and X.C. designed the research and developed the high-speed stress microscopy. T.-P.S. performed the impact stress measurements. T.-P.S., L.G., and X.C. discussed and analyzed experimental data. L.G. solved the numerical solution of drop impact on elastic media with input from X.C. F.A.-N., K.A., P.G., and L.G. studied drop-impact erosion on plaster slabs. X.C. conceived and supervised the project. T.-P.S. and X.C. cowrote the manuscript. All authors discussed and commented on the manuscript.

## Competing interests

The authors declare no competing interests.
