## [Peer Review File · Nature Communications]

Title: Stress distribution and surface shock wave of drop impactREVIEWER COMMENTS

Reviewer #1 (Remarks to the Author):

The authors present a new experimental method to indirectly measure stress and pressure distribution in a deformable substrate, tracking fluorescent particles.

This is a very interesting paper, proposing a novel technique and I support publication of the paper. Nonetheless, I have some doubts that I encourage the authors to address to improve some key aspects of the paper.

Major comments:

- page 4. The authors claim “Experiments and simulations match quantitatively, verifying the accuracy of high-speed stress microscopy (Fig. 2b)”. However, there is no description in the main text of how DIC correlation is used to extract shear stress and pressure distribution from tracking particle motion (i.e. substrate deformation). I found details in the Materials and Methods, but since the authors are presenting a new method, it should also be discussed in the main text. The method is far from trivial and the authors should describe how the method works, before making a strong statement on the matching between experiments and simulations.
- Methods section. The authors claim: “Instead, we adopted the quasi-steady state assumption to calculate the pressure”. It’s not clear to me how a with a quasi-state assumption is justified, even in strongly dynamic conditions.
- In the drop impact community, we have observed a long-lasting debate on whether the maximum pressure at impact scales as the kinetic pressure $\rho \cdot D \cdot V^2$, or as the water-hammer pressure $\rho \cdot c \cdot V$. This is for example important to predict the resistance to impalement in superhydrophobic surfaces. I suggest the authors to provide a discussion the pressure scaling as function of velocity.
- The authors should make sure they cite other studies, which has been published recently, using e.g. traction force microscopy in the context of wetting. One ex.: Gerber, J., Lendenmann, T., Eghlidi, H. et al. Wetting transitions in droplet drying on soft materials. Nat Commun 10, 4776 (2019). <https://doi.org/10.1038/s41467-019-12093-w>

Reviewer #3 (Remarks to the Author):

The current work addresses an important area of tribology, i.e. impact erosion due to droplets. Rain erosion of e.g. the leading edge of wind turbine blades is an application area where much work is conducted recently. Important pioneering work on the physics of droplet impact is and was conducted by J.E. Fields, Cambridge university. This reviewer suggests 1) replacing words like the ‘mysterious origin’ (abstract) and references to Laozi, by references to ongoing work in engineering technology and validate the results in the context of the work of J.E. Fields and e.g. W.F. Adler or other results on droplet impact that are available in literature. This would also be in line with the reference to the Rayleigh wave on page 7. Key aspect in tribology is the systems dependence of wear. The advanced and innovative

method that is presented in this work, accompanied by a thorough fluid mechanical analysis, makes however use of academic model materials. That 2) raises the question on how well these results can be translated into the erosion of engineering surfaces and engineering materials that range from soft materials like rubber to steels to ceramics and have roughness and imperfections.

Reviewer #4 (Remarks to the Author):

Please see attached review.

Referee report on:

“Erosion by dripping drops: The stress distribution and surface shock wave of drop impact”

by Sun, Alvarez-Novoa, Andrade, Gutierrez, Gordillo & Cheng.

This manuscript describes experiments and simulations of the problem of drop-impact erosion. The work uses a substrate gel seeded with micro-particles to measure the deformation of the gel. The work is novel and addresses an important open question. While there are certain drawbacks to the study, the work opens up a new possibilities for experimentation which I think will be exploited in the future. I therefore recommend the study be published in *Nature Communications*, if the following points are adequately addressed.

- 1. The work of Howland *et al.* “It’s Harder to Splash on Soft Solids” *Phys. Rev. Lett.*, **117**, 184502 (2016), is quite relevant and should be cited. Their figure 4 shows how the maximum pressure moves away from the centerline during the impact. How does their result in figure 4 compare to the present Figures 4(c)? Furthermore, this paper shows clearly that the stiffness of the substrate greatly affects the lamellar dynamics. The statement in the Supplemental Information “*indicating the weak effect of the gel stiffness on the kinematics of impacting drops within the range of our experiments*”, therefore needs to be better supported with reference to this paper.
- 2. The air-cushioning is mentioned on page 9 and discussed in the Discussion section. This can be done more clearly, with the most up-to-date references. The 2-D theory and simulation of Mandre *et al.* *Phys. Rev. Lett.* (2009) shows that the largest pressure is at the “*kink*” outside the central air-disc. This kink is not mentioned in the text, but I feel is quite important. The best time-resolved imaging of this effect should also be cited. Li & Thoroddsen *J. Fluid Mech.* (2015), have reported measurements with $0.2 \mu\text{s}$ time resolution, which is more than two orders of magnitude faster than used in this manuscript. Both of these papers discuss the radial speed of the kink, which can be 50 times the impact velocity.

The compressibility of the gas can come into play, as first measured

by Liu *et al.* “Compressible air entrapment in high-speed drop impacts on solid surfaces”. *J. Fluid Mech.* **716**, R9 (2013). The maximum compression of the air occurs during the first micro-seconds, so time-resolution of 25 μs would have a hard time of catching this? Do the authors expect this to show up in the shear stress? The earliest motions of the turning point, have also been measured with 1 μs time-resolution in figure 7 of Zhang *et al.* *J. Fluid Mech.* **883**, A46 (2020).

- 3. The Discussion also mentions doing experiments under reduced pressure. This has already been done, showing much smaller contact air-disc, which would contribute to a larger initial “supersonic” regime in Fig. 7(b), see Li *et al.* *Phys. Rev. Lett.* “Double Contact During Drop Impact on a Solid Under Reduced Air Pressure”, **119**, 214502 (2017).
- 4. In other words, please put the above three items in better context in the Discussion or Introduction sections. There are much high-resolution measurements available than the ones you cite.
- 5. The captions of Figures 2, 3 etc. should include more information about the impacts, especially U and D . It is not sufficient to simply refer to the text.
- 6. The very soft substrates used in the experiments, may give quite different results from typical solid substrates, like stone. The jump to infinite rigidity is perhaps too large a jump here. What about using stiffnesses for typical solid surfaces, like glass or acrylic? I wondered this during the early part of the work, but then it is mentioned in the Discussion section. I suggested also referring to this earlier in the manuscript.
- 7. The colors in Figure 6, are two orders of magnitude stronger for the solid impact (kPa vs Pa) than for the drop, so comparison is a little misleading. Perhaps mention this in the caption.
- 8. The comparison between experiment and simulation in Fig. 2, are similar, but I would not say “match”. Why are the colors inverted between the two panels in Fig. 2b?
- 9. The numerical simulations do not appear to be very well resolved. For example is the air-cushioning during the earliest stage captured? This needs to be better described.

- 10. With 120% weight of the salt compared to that of the water, it is surprising that neither the surface tension of the viscosity changes. Was this measured or found in the literature? Please clarify.
- 11. On page 4 it is stated that locations are non-dimensionalized by the drop diameter D . Does this apply for the various radii discussed? Please make this clear.
- 12. Finally, the arguments in the Discussion section seem to suggest that the drop is more efficient in cracking the substrate than the solid. Do the authors seriously argue that repeated impacts of solid spheres will do less damage? This needs to be better described.
- Minor Issues:
 - i. The inset in Fig. S2(a) is confusing. It would be better to make a separate panel for this.
 - ii. Please use the same extent of the x -axis of Fig. 7(a) and (c), to allow easier comparison. There does not seem to be useful information on the right-most side.
 - iii. Perhaps the Discussion section should be Discussion and Conclusions?
 - iv. The second line in the abstract is over-hyped: “calling for tremendous prevention and preservation efforts”. I would tone this down.
 - v. Page 7-8: strong sharp surface wave \rightarrow strong and sharp surface wave ?
 - vi. Where is Fig. 1(b) taken from? The image of the PDMS in Fig. 2(a) should be larger.

We report below a detailed response to the referees' reports. The original referee comments are in *italic*, and our response appears in normal font. When addressing the referees' concerns, we also explicitly listed all the changes we made in the revised manuscript. All the changes are also highlighted in red in the revised manuscript.

Response to Reviewer 1:

We thank Reviewer 1 for the recommendation of the publication of our work and for the constructive questions, which helped us to strengthen further the conclusion of our study and greatly improve the readability of the manuscript. Below, we will address the comments/concerns of the reviewer in detail:

1. page 4. The authors claim "Experiments and simulations match quantitatively, verifying the accuracy of high-speed stress microscopy (Fig. 2b)". However, there is no description in the main text of how DIC correlation is used to extract shear stress and pressure distribution from tracking particle motion (i.e. substrate deformation). I found details in the Materials and Methods, but since the authors are presenting a new method, it should also be discussed in the main text. The method is far from trivial and the authors should describe how the method works, before making a strong statement on the matching between experiments and simulations.

We fully agree with the reviewer. Since one of the most important contributions of our manuscript is the development of the new experimental technique, we should discuss the technique in detail in the main text, instead of in the Materials and Method. Following the suggestion of the reviewer, we have moved the discussion of the experimental method from the Materials and Methods to the main text. In particular, the detailed method on how to calculate the stress distributions from the DIC correlation is now presented in the revised main text.

We highlighted the changes in red in the revised main text. Please see the revised manuscript for details.

2. Methods section. The authors claim: "Instead, we adopted the quasi-steady state assumption to calculate the pressure". It's not clear to me how a with a quasi-state assumption is justified, even in strongly dynamic conditions.

The quasi-steady-state assumption applies when the elastic force from the deformation of the elastic gel dominates the inertial force from the acceleration of the gel displacement. We have calculated the ratio between the elastic force and the inertial force of the gel from our experiments. The ratio is less than $\sim 2\%$ in the region near the impact axis interested in our research, justifying the application of the assumption.

We have also verified the assumption directly by comparing our experimental measurements on the impact pressure of solid-sphere impact with the numerical results from finite element simulations. The pressure distribution of solid-sphere impact can be well captured by standard finite element simulations, which provide a quantitative reference to verify our experimental technique. As shown in Fig. 2b, our measurements based on the quasi-steady-state assumption quantitatively agree with the numerical results, which again justifies the validity of the assumption. As the time scale of solid-sphere impact is much shorter than that of drop impact under comparable impact conditions, the quasi-steady-state assumption should work even better for drop impact.

Lastly, our paper focuses on the quantitative features of the shear stress of drop impact as the leading cause of surface erosion, which does not depend on the quasi-steady-state assumption.

To address the question of the reviewer in the revision, we have added the following sentence in pg. 4 of the revised main text:

“We have verified the assumption by comparing the inertial force and the elastic force in the impact process (SI Sec. 2) and by comparing experimental and numerical results on the impact pressure of solid-sphere impact (see below).”

Furthermore, we have discussed the details of the verification at the end of Sec. 2 of the Supplementary Information.

3. In the drop impact community, we have observed a long-lasting debate on whether the maximum pressure at impact scales as the kinetic pressure $\rho D V^2$, or as the water-hammer pressure $\rho c V$. This is for example important to predict the resistance to impalement in superhydrophobic surfaces. I suggest the authors to provide a discussion the pressure scaling as function of velocity.

The kinetic pressure (or the inertial pressure) scales as ρU^2 , instead of $\rho D U^2$, which we assume is a typo of the report.

The water-hammer pressure is induced by the compression wave, which occurs over a very short time interval of $t_w \sim DU/c^2$ at the beginning of drop impact when the contact diameter between the drop and the substrate is $D_w \sim DU/c$ [see Ref. 20 (Soto et al, Soft Matter 2014)]. Here, D is the diameter and U is the impact velocity of the drop. c is the speed of sound in the liquid. $t_w \sim 5$ ns for drop impact of our experiments, which is more than three orders of magnitude smaller than the temporal resolution of our experiments. $D_w \sim 7$ μm , which is more than one order of magnitude smaller than the spatial resolution of our experiments. Thus, the pressure and shear stress measured in our study must be kinetic in origin. Consistent with the argument, our previous study on the impact force of liquid drops under the similar impact condition has shown that the maximum impact force scales with the kinetic pressure, $\rho U^2 D^2$, instead of the water-hammer pressure $\rho c U D^2$ (Ref. 23). We have discussed in detail about the two different velocity scalings there. In addition, Ref. 7 and 20 also provided nice discussions on why the water hammer pressure is hardly relevant to drop impact at relatively low impact velocities, which is considered here in this study.

Following the suggestion of the reviewer, we have added the following discussion in pg. 6 of the revised main text:

“Here, $\tau(r,t)$ is non-dimensionalized by the inertial pressure ρU^2 . Note that the much stronger water-hammer pressure $\rho U c$ associated with the compression wave occurs on the time scale of a few nanoseconds, which is too short to be relevant in our current experiments^{7,20,23}.”

4. The authors should make sure they cite other studies, which has been published recently, using e.g. traction force microscopy in the context of wetting. One ex.: Gerber, J., Lendenmann, T., Eghlidi, H. et al. Wetting transitions in droplet drying on soft materials. Nat Commun 10, 4776 (2019). <https://doi.org/10.1038/s41467-019-12093-w>

We thank the reviewer for providing us with this very interesting reference. We have now added the new reference in the revised manuscript (Ref. 29).

Response to Reviewer 3:

We thank the reviewer for providing us the useful references and suggestions. We have revised the manuscript in response to these suggestions/comments below.

The current work addresses an important area of tribology, i.e. impact erosion due to droplets. Rain erosion of e.g. the leading edge of wind turbine blades is an application area where much work is conducted recently. Important pioneering work on the physics of droplet impact is and was conducted by J.E. Fields, Cambridge university. This reviewer suggests 1) replacing words like the 'mysterious origin' (abstract) and references to Laozi, by references to ongoing work in engineering technology and validate the results in the context of the work of J.E. Fields and e.g. W.F. Adler or other results on droplet impact that are available in literature. This would also be in line with the reference to the Rayleigh wave on page 7.

We thank the reviewer for bringing our attention to the pioneering work of the late J. E. Field, who has sadly passed away last year. His work has been nicely summarized in his own review article (Lesser, M. B. & Field, J. E. “The impact of *compressible* liquids”, *Annu. Rev. Fluid Mech.* 15, 97–122 (1983)). Following the suggestion, we have cited the review as well as the work of W. F. Alder as Ref. 13 and 14 of our revised manuscript.

We would like to point out that these pioneering studies in 1970s and 80s focused on the drop impact of *compressible* liquids with the impact velocity U at a few hundred meters per second and Mach number U/c on the order of one. Here, c is the speed of sound in liquids. In contrast, our manuscript investigated the drop impact of *incompressible* drops with U on the order 1 to 10 m/s, which are more relevant to natural raindrop impact and most environmental and engineering processes. (The terminal velocity of raindrops is about 10 m/s and the drop speed in inkjet printers is about 2 to 20 m/s.) Due to its ubiquity and fundamental interests, the impact of incompressible drops at relatively low impact speeds has dominated the study of drop impact in recent years in the physics and fluid mechanics communities (see the highly cited reviews Ref. 16 and 17 and references therein). It has been well established now that the dynamics of drop impact is very different in nature in these two different impact processes (see e.g. the discussion in Ref. 7, 16, 20 and 42). Particularly, the mechanisms underlying drop-impact erosion are qualitatively different in the two processes. While the impact erosion of compressible drops is through the surface fracture caused by the strong water-hammer pressure of high-speed drops on the order of $\rho c U$, the erosion of incompressible drop impact at low impact speeds is induced by the much weaker inertial pressure/shear stress on the order of ρU^2 . Here, ρ is the density of liquids. (Please see our answer to Question 3 of Reviewer 1 for the further discussion of the two qualitatively different pressure scales.) Although the impact erosion of high-speed compressible drops has been extensively studied by Field, Adler and many others, it is still far from clear what is the physical origin of the impact erosion of low-speed incompressible drops, which is the focus of our current study. We believe Laozi's saying on dripping water drops from the 5th-century BC refers to low-speed

incompressible drop impact, instead of high-speed compressible drop impact associated with modern technologies. His saying motivates our research.

Following the suggestion of the reviewer, we have now removed the word “mysterious” and also emphasized our study on low-speed drop impact in the revised abstract:

“Our study opens the door for quantitative measurements of the impact stress of liquid drops and sheds light on the origin of *low-speed* drop-impact erosion.”

In addition, we have now emphasized the difference between the impact of high-speed compressible drops studied by Field and Alder and the impact of low-speed incompressible drops explored in our study in the Introduction of the revised manuscript:

“While the impact damage on solid substrates caused by high-speed compressible liquid drops with the impact velocity on the order of a few hundred meters per second has been well explored^{13,14}, our understanding of the impact erosion of *low-speed incompressible* drops relevant to most natural and engineering processes is still rudimentary.”

Key aspect in tribology is the systems dependence of wear. The advanced and innovative method that is presented in this work, accompanied by a thorough fluid mechanical analysis, makes however use of academic model materials. That 2) raises the question on how well these results can be translated into the erosion of engineering surfaces and engineering materials that range from soft materials like rubber to steels to ceramics and have roughness and imperfections.

We are very pleased to see the reviewer’s positive comments on our study. Our study showed the first measurements on the pressure and shear stress distributions of drop impact on smooth elastic substrates. The experimental technique developed in our study can be easily extended to study the stress of drop impact in more complicated situations such as drop impact on patterned substrates (with different roughness and imperfections), at reduced ambient pressure and with non-Newtonian drops. Thus, our fundamental study provides not only the basis of reference in knowledge for understanding the impact stress on more complicated material substrates, but also a crucial technical platform for future experimental investigations on different material systems. This view is shared by the other reviewers of the manuscript. They have commented that our “novel technique” “opens up new possibilities for experimentation”, which “will be exploited in the future.”

Response to Reviewer 4:

We thank the reviewer for carefully reading our manuscript and for recommending the publication of our work. We also greatly appreciate the thorough and very informative suggestions/comments of the reviewer, which helped us significantly improve the quality of our manuscript. Below, we will address the questions of the reviewer in detail:

1. The work of Howland et al. "It's Harder to Splash on Soft Solids" Phys. Rev. Lett., 117, 184502 (2016), is quite relevant and should be cited. Their figure 4 shows how the maximum pressure moves away from the centerline during the impact. How does their result in figure 4 compare to the present Figures 4(c)? Furthermore, this paper shows clearly that the stiffness of the substrate greatly affects the lamellar dynamics. The statement in the Supplemental Information "indicating the weak effect of the gel stiffness on the kinematics of impacting drops within the range of our experiments", therefore needs to be better supported with reference to this paper.

Thanks for the reviewer for providing us this important reference, which was missing in our original manuscript. We have added the reference as Ref. 37 in the revised manuscript.

Figure 4 of Ref. 37 shows the maximum pressure as a function of time, $p_{max}(t)$, whereas Figure 4c of our manuscript shows the shear stress as a function of the radial position, $\tau(r)$. It is hard to compare the normal stress with the shear stress directly. Nevertheless, Figure 6b of our manuscript shows the propagation of the maximum pressure away from the centerline, which qualitatively agrees with the numerical results of Figure 4 in Ref. 37.

We agree with the reviewer that our claim on the weak effect of the gel stiffness on the kinematics of impacting drops in the SI is not well founded. Our experiments showed that the position of the maximum shear stress does not depend on the stiffness of the elastic gel in the range of our experiments from $E = 100$ kPa up to $E = 420$ kPa (Fig. 4b). The result, however, does not necessarily indicate the weak effect of the gel stiffness on other kinematic features of impacting drops such as the motions of lamella. To be more accurate, we have removed this claim from the SI, which does not affect the conclusion of our manuscript.

2. The air-cushioning is mentioned on page 9 and discussed in the Discussion section. This can be done more clearly, with the most up-to-date references. The 2-D theory and simulation of Mandre et al. Phys. Rev. Lett. (2009) shows that the largest pressure is at the "kink" outside the central air-disc. This kink is not mentioned in the text, but I feel is quite important. The best time-resolved imaging of this effect should also be cited. Li & Thoroddsen J. Fluid Mech. (2015), have reported measurements with $0.2 \mu s$ time resolution, which is more than two orders of magnitude faster than used in this manuscript. Both of these papers discuss the radial speed of the kink, which can be 50 times the impact velocity.

Although we have cited Mandre et al PRL (2009), we did not discuss the kink structure in the original manuscript. We agree with the reviewer that the kink structure is the key feature associated with the fast radially propagating pressure maximum. To address the issue, we have now added the discussion of the kink structure, its relation to the maximum pressure, as well as its fast propagating speed in the Discussion section of the revised manuscript. The interesting work of Li & Thoroddsen has also been added as a new reference (Ref. 44) in the revised manuscript.

The revised manuscript reads as:

“Particularly, the underlying air layer induces a micron or sub-micron dimple-like deformation bounded by a kink structure at the bottom of an impacting drop^{41,44}. The pressure reaches the maximum underneath the kink^{41,42}, which propagates radially outwards at a speed 50 times higher than the impact velocity at early times of $t \sim 10^{-4}$.^{41,44}”

The compressibility of the gas can come into play, as first measured by Liu et al. “Compressible air entrapment in high-speed drop impacts on solid surfaces”. J. Fluid Mech. 716, R9 (2013). The maximum compression of the air occurs during the first micro-seconds, so time-resolution of 25 μ s would have a hard time of catching this? Do the authors expect this to show up in the shear stress? The earliest motions of the turning point, have also been measured with 1 μ s time-resolution in figure 7 of Zhang et al. J. Fluid Mech. 883, A46 (2020).

Unfortunately, we are not able to achieve a higher temporal resolution in our current experiments. The limitation is mainly because of the hardware limit of our high-speed camera, as well as the small number photons emitted by fluorescent tracer particles used in our study. We do hypothesize that air compressibility demonstrated beautifully by Liu and co-workers would also lead to interesting features in shear stress on the impacted surface. It would be an interesting but certainly challenging direction for future experimental studies. To the best of our knowledge, there is no theoretical or numerical works probing the shear stress at very early times of drop impact either.

We have replaced the previous reference with the new reference of Zhang et al. JFM (2020) in our revised manuscript (Ref. 38), which shows the square-root scaling of the turning point at even earlier times and therefore further strengthens the conclusion of our work.

3. The Discussion also mentions doing experiments under reduced pressure. This has already been done, showing much smaller contact air-disc, which would contribute to a larger initial “supersonic” regime in Fig. 7(b), see Li et al. Phys. Rev. Lett. “Double Contact During Drop Impact on a Solid Under Reduced Air Pressure”, 119, 214502 (2017).

We have added Li et al PRL (2017) as a new reference of the manuscript (Ref. 45) and discussed the effect of reduced ambient pressure in the Discussion of the revised manuscript.

“The radius and the thickness of the air dimple become significantly smaller at reduced ambient pressure⁴⁵, which eventually leads to the vanishing of air cushioning and the recovery of the singular dynamics of the contact line at short times.”

4. In other words, please put the above three items in better context in the Discussion or Introduction sections. There are much high-resolution measurements available than the ones you cite.

We greatly appreciate the reviewer for providing us all the new references, particularly the series of important experimental works from the Thoroddsen group on the early-time dynamics of impacting drops. The new references greatly strengthen the conclusion of our manuscript.

5. The captions of Figures 2, 3 etc. should include more information about the impacts, especially U and D . It is not sufficient to simply refer to the text.

We have added the value of U and D in the caption of Figs. 2-7.

6. The very soft substrates used in the experiments, may give quite different results from typical solid substrates, like stone. The jump to infinite rigidity is perhaps too large a jump here. What about using stiffnesses for typical solid surfaces, like glass or acrylic? I wondered this during the early part of the work, but then it is mentioned in the Discussion section. I suggested also referring to this earlier in the manuscript.

We cannot use very stiff gels for the experiments, as the deformation of the gels would be too small to measure accurately by DIC. Following the suggestion of the reviewer, we have added the following sentence in the early part of the revised manuscript in the Experiment section:

“The effect of the stiffness of the impacted surface shall be further discussed in the Discussion section.”

7. The colors in Figure 6, are two orders of magnitude stronger for the solid impact (kPa vs Pa) than for the drop, so comparison is a little misleading. Perhaps mention this in the caption.

We have added the following comment in the caption of Fig. 6:

“Note that the pressure scale is kPa for the solid-sphere impact and Pa for the drop impact.”

8. The comparison between experiment and simulation in Fig. 2, are similar, but I would not say “match”. Why are the colors inverted between the two panels in Fig. 2b?

We have modified the sentence to:

“Experimental measurements agree well with the numerical results, verifying the accuracy of high-speed stress microscopy (Fig. 2b).”

We defined the normal pressure on the substrate to be negative in the original figure 2b, which was represented by blue color. We have now revised the color code of the lower panel of Fig. 2b, so that the pressure is positive and the color codes are consistent between the two panels.

9. The numerical simulations do not appear to be very well resolved. For example is the air-cushioning during the earliest stage captured? This needs to be better described.

We took the theoretical formulas of the surface pressure and shear stress of incompressible drops on a solid substrate from Ref. 39 as the boundary condition of our simulation, which do not include the effect of air cushioning. Our current study focused on experiments. Simulations were mainly used to calibrate our experimental technique and qualitatively support the finding of our experiments. More advanced simulations are beyond the current capability of our group. We have to leave the more accurate numerical study to research groups that are professional in the numerical study of drop impact.

To remove any ambiguity, we have explicitly commented this point in Sec. 3 of the revised SI, where we discussed the detail of the simulation:

“Under the one-way approximation, we apply the pressure and shear distributions of incompressible drops on infinitely rigid substrates **without the effect of ambient air** as the boundary conditions of Eqs. (12) and (13)”

10. With 120% weight of the salt compared to that of the water, it is surprising that neither the surface tension of the viscosity changes. Was this measured or found in the literature? Please clarify.

The viscosity of the NaI solution is obtained from literature (Abdulagatov, et al. J. Chem. Eng. Data **51**, 1645 (2006)), which shows the viscosity of the 60% w/w solution at 25 °C is 1.12 mPa·s (Fig. 1 of the paper). We estimated the surface tension of the solution based on the contact angle of the solution on untreated microscope coverslips. The contact angle is the same as that of pure water within the experimental errors.

To address the concern of the reviewer, we have used the more accurate value of the viscosity of the solution in calculating the Reynolds number and cited the above literature (Ref. 31).

In addition, we have added the following comment regarding surface tension:

“The surface tension of the solution σ is similar to that of water based on contact angle measurements.”

11. On page 4 it is stated that locations are non-dimensionalized by the drop diameter D . Does this apply for the various radii discussed? Please make this clear.

Yes, we non-dimensionalized the locations using the diameter of the drop or the solid sphere of the corresponding experiments.

To clarify the point, we revised the text as follows:

“Positions and times are reported in the dimensionless forms using D and D/U of the corresponding experiments as the length and time scale, respectively.”

12. Finally, the arguments in the Discussion section seem to suggest that the drop is more efficient in cracking the substrate than the solid. Do the authors seriously argue that repeated impacts of solid spheres will do less damage? This needs to be better described.

Thanks for pointing out the confusing writing of our original manuscript. We certainly do not argue solid-sphere impact is less damaging than drop impact. Rather, we would like to illustrate the different nature of drop-impact erosion.

To remove the confusion, we have revised the first sentence of the Discussion section as follows:

“Taken together, our high-speed stress microscopy reveals three unique dynamic features of drop impact. These features contribute to the ability of drop impact to erode solid surfaces and result in the distinct nature of drop-impact erosion, qualitatively different from that of solid-sphere-impact.”

Minor Issues:

i. The inset in Fig. S2(a) is confusing. It would be better to make a separate panel for this.

As the propagation of the weak bulk elastic wave is irrelevant to the main point of our manuscript. We have completely removed the inset of Fig. S2(a).

ii. Please use the same extent of the x-axis of Fig. 7(a) and (c), to allow easier comparison. There does not seem to be useful information on the right-most side.

Changed. Note that the maximum pressure propagates with the turning point in our simulations, which is faster than that observed in experiments. Experiments and simulations show qualitative similarities.

iii. Perhaps the Discussion section should be Discussion and Conclusions?

Changed.

iv. The second line in the abstract is over-hyped: “calling for tremendous prevention and preservation efforts”. I would tone this down.

We have modified the sentence as follows:

“..., calling for substantial prevention and preservation efforts.”

v. Page 7-8: strong sharp surface wave → strong and sharp surface wave?

Changed.

vi. Where is Fig. 1(b) taken from? The image of the PDMS in Fig. 2(a) should be larger.

Figure 1b is from our own experiments. The image of the PDMS has been enlarged in the revised Fig. 2a.

REVIEWER COMMENTS

Reviewer #1 (Remarks to the Author):

The authors have addressed my comments. The paper can thus be published.

Reviewer #4 (Remarks to the Author):

Please see attached file.

Referee report on Revised Manuscript NCOMMS-21-33181A:

”Erosion by dripping drops:

The stress distribution and surface shock wave of drop impact”

by Sun, Alvarez-Novoa, Andrade, Gutierrez, Gordillo & Cheng

I have read carefully through the authors’ reply to my earlier concerns and suggestions. They have made excellent effort of fixing these points and I can now support publication, with one more suggestion.

- The only additional comment would be regarding their reply to my previous question 10: on the effect of the high concentration of the salt on the surface tension. Their reply:

“The surface tension of the solution σ is similar to that of water based on contact angle measurements.”

is not entirely satisfactory. Was this done with very small droplets? The greatly added density, compared to water, will certainly reduce the capillary length, making the drop deform more by gravity. Furthermore, the addition of the salt may change the “surface energy” between the liquid and the solid, thereby altering the contact angle, irrespective of the liquid-gas surface tension. It would be more convincing to simply measure this with a tensiometer, if it is at all possible.

As Reviewer #1 supports the publication of our manuscript without any further questions, we report below only a detailed response to Reviewer #4. The original reviewer comment is in *italic*, and our response is in normal font. We explicitly state the changes we have made in the revision at the end. These changes have also been highlighted in red in the revised text.

Response to Reviewer #4:

We thank Reviewer #4 for re-reading our manuscript and for recommending the publication of our manuscript. Below, we address the one remaining comment of the reviewer.

The only additional comment would be regarding their reply to my previous question 10: on the effect of the high concentration of the salt on the surface tension. Their reply:

“The surface tension of the solution σ is similar to that of water based on contact angle measurements.”

is not entirely satisfactory. Was this done with very small droplets? The greatly added density, compared to water, will certainly reduce the capillary length, making the drop deform more by gravity. Furthermore, the addition of the salt may change the “surface energy” between the liquid and the solid, thereby altering the contact angle, irrespective of the liquid-gas surface tension. It would be more convincing to simply measure this with a tensiometer, if it is at all possible.

We agree with the reviewer that contact angle measurements provide only a crude estimate of surface tension. Indeed, the surface tension of electrolyte solutions including that of NaI solutions has been well documented and measured using pendant-drop tensiometry. The study by Chen *et al.* (*J. Chem. Eng. Data* **62**, 3783-3792 (2017)) shows that the surface tension of NaI solution at 60% w/w is 81.3 mN/m at 25 °C (with a minor extrapolation), which is an 11% increase compared with the surface tension of water at 71.97 mN/m at the same temperature.

Practically, the only reason we need to know the surface tension of drops in our work is to make sure the Weber number (We) of impact is much larger than one. Using the more accurate value from tensiometry, we have $We = 833$, instead of 940 in our original manuscript, which still satisfies $We \gg 1$ and therefore justifies the conclusion that our drop impact is dominated by fluid inertia at early times. In fact, to have $We = 1$, the surface tension of liquids needs to be 67.7 N/m, 940 times larger than that of water, which is unphysical. Hence, although the value of the surface tension of NaI solutions might not be accurate, it does not affect any conclusion of our manuscript.

To address the concern of the reviewer, we have used the more accurate value of the surface tension of the NaI solution from tensiometry for the We calculation and cited the study above as a new reference (Ref. 32) in our revised manuscript:

“The surface tension of the solution $\sigma \approx 81.3$ mN/m, slightly larger than that of water based on a pendant-drop tensiometry measurement³².”

“..., yielding a Reynolds number $Re = \rho UD/\eta = 20360$ and a Weber number $We = \rho DU^2/\sigma = 833$.”